# Bilateral Deficits in Dynamic Postural Stability in Females Persist Years after Unilateral ACL Injury and Are Modulated by the Match between Injury Side and Leg Dominance

**DOI:** 10.3390/brainsci13121721

**Published:** 2023-12-16

**Authors:** Maité Calisti, Maurice Mohr, Peter Federolf

**Affiliations:** Department of Sport Science, University of Innsbruck, 6020 Innsbruck, Austria; maite.calisti@uibk.ac.at (M.C.); maurice.mohr@uibk.ac.at (M.M.)

**Keywords:** ACL injuries, neuromuscular control deficits, postural control, rehabilitation, leg dominance, lateralization, postural stability, jump-landing, re-injury

## Abstract

Previous research has documented brain plasticity after an anterior cruciate ligament (ACL) tear and suggests that these neural adaptations contribute to poorer motor control. Since both brain hemispheres show adaptations, we hypothesized that reduced dynamic stability occurs not only in the injured, but also the contralateral, uninjured leg. Further, given brain hemispheric specialization’s impact on motor coordination, we hypothesized the need to consider the injury side. A total of 41 female athletes and 18 controls performed single-leg jump-landings. Dynamic postural stability was measured as time-to-stabilization (TTS). We found reduced medio-lateral dynamic stability for the ACL injured leg (*p* = 0.006) with a similar trend for the contralateral leg (*p* = 0.050) compared to the control group. However, when distinguishing between injuries to the dominant and non-dominant legs, we found increased medio-lateral TTS only if the injury had occurred on the dominant side where landings on injured (*p* = 0.006) and contralateral (*p* = 0.036) legs required increased TTS. Assessments of dynamic stability, e.g., in the context of return-to-sport, should consider the injury side and compare results not only between the injured and the contralateral leg, but also to uninjured controls. Future research should not pool data from the dominant-leg ACL with non-dominant-leg ACL injuries when assessing post-injury motor performance.

## 1. Introduction

More than 20% of individuals who have undergone primary anterior cruciate ligament (ACL) reconstruction (ACLR) will suffer a secondary ACL injury, either as a recurrence to the already injured ipsilateral or to the contralateral leg [1]. In fact, the risk of sustaining a new rupture to the healthy contralateral ACL has been reported to be significantly higher than the risk of a re-rupture of the reconstructed ACL [2].

Poor neuromuscular control following primary ACL rupture, e.g., through inappropriate preparatory and reactive muscle activity [3], has been suggested as one contributing factor to the increased risk of re-injury [4]. Neuromuscular control deficits may be innate to the individual, but can also arise from the primary ACL rupture, specifically from the loss of afferent information due to severed neuroreceptors in the torn ACL [5]. This disruption in the afferent input from the injured knee joint to the central nervous system (CNS) may lead to the reorganization of central nervous processes involved in controlling the knee joint movement and stability [4,5]. Importantly, this reorganization has been observed in both brain hemispheres and for sensory processes related to both the previously injured ipsilateral knee and the healthy contralateral knee [6]. For example, Grooms et al. [6] investigated fMRI-based brain activity during a knee flexion–extension task and demonstrated that following unilateral ACLR individuals showed greater activity in brain regions responsible for the processing of sensory information from both the injured and contralateral leg compared to a control group. As a result, it can be hypothesized that a unilateral ACLR will alter the neuromuscular control bilaterally, which could explain the high re-injury rates for both knees after unilateral ACLR. Further, if neural reorganization due to an ACL injury is a cause of neuromuscular control deficits, and given that the well-established hemispheric specialization of the brain [7] affects motor coordination and performance [8], we hypothesize that the location of the injury (i.e., on the dominant or non-dominant side) may affect the motor control function. This, in turn, would suggest a role for the injury side and that laterality could potentially affect stability and injury or reinjury risk, as documented by Negrete et al. [9].

Previous studies lend initial support to the neural mechanisms contributing to high rates of contralateral ACL injuries. For example, Culvenor et al. [10] demonstrated bilateral proprioceptive deficits following ACLR when performing a single-leg squat. When testing knee joint proprioception, Reider et al. [11] found bilateral deficits in the knee joint proprioception after a unilateral ACL injury. However, assessing neuromuscular control deficits in a static and single-leg stance position may not sufficiently challenge the neuromuscular control system to indicate tangible effects on reinjury risks. Instead, jump-landing tasks or changes of direction may be more suitable for assessing dynamic postural stability relevant to sports and functional activities. “Time to stabilization” (TTS) is a functional outcome measure of dynamic postural stability by assessing the time it takes for participants to achieve a static position after a jump-landing task [12]. TTS is based on the orthogonal components of the ground reaction force (GRF) vector obtained by a force plate [13].

One of the first studies by Colby et al. [14] examined the TTS in ACL reconstructed legs in comparison to the contralateral leg. They found that the reconstructed knee showed greater TTS in the vertical force direction compared to the uninjured leg during a step-down task [14]. Webster and Gribble [12] examined the resultant vector of the TTS in ACLR knees compared to healthy knees and found that ACLR athletes took significantly longer to stabilize after a jump. Duprey et al. [15] also found longer TTS times in the vertical direction during a backward landing in individuals who went on to sustain an ACL rupture in the future. However, there is limited evidence regarding dynamic postural stability for the healthy contralateral leg in people with a previous unilateral ACLR. Further, we are not aware of any study that has considered leg dominance as a factor for TTS stabilization results after ACLRs.

Therefore, the objective of this study was to assess neuromuscular control deficits following unilateral ACL injury through the TTS during single-leg jump-landing tasks on both legs and in comparison to a control group. Although many studies used the vertical ground reaction force component to calculate the TTS, we additionally focused on anterior–posterior as well as medio–lateral force components as knee injury risk factors are often related to medio–lateral instability [16]. Following the mechanism described above, we tested the primary hypothesis (H1) that dynamic postural stability as quantified by TTS would be lower for both the healthy contralateral leg and the ACL injured leg compared to a control group with no knee injury history. In addition, we also tested the secondary hypothesis (H2) that the match between leg dominance and the side of the injury would impact dynamic postural stability in the ACL group.

## 2. Materials and Methods

### 2.1. Participants

A convenience sample of 43 female university students participated in this cross-sectional study. A total of 23 participants had previously sustained an ACL injury, while 20 participants were free of ACL injuries (Table 1). A post hoc sensitivity analysis with a significance level of alpha = 0.05 indicated that this sample size enabled us to detect between-group effects (ACL vs. control group) of moderate to large effect sizes (Cohen’s d > 0.77) with a desired power of 0.8. Only female athletes were recruited because, compared to males, they are at two to four times greater risk of a primary ACL injury [17]. Leg dominance was defined as the preferred kicking leg [18]. Inclusion criteria for both groups were (1) absence of any lower extremity injury in the previous 6 months, (2) absence of any previous severe ankle injury, (3) familiarity with jump-landing-tasks due to participation in respective sports. Additional inclusion criteria for the ACL group were (1) one or two ACL injuries to the ipsilateral leg (same leg), (2) clearance to return to level I sports (i.e., sports that involve jumping, hard pivoting and cutting) for at least 1 year before participation in the current study. Participants were recruited via the university email list and personal contact. The study was conducted between January 2020 and December 2021. Beforehand, the study had been approved by the local ethics board of the University of Innsbruck (certificate 71/2019). All procedures of the study were conducted in accordance with the ethical principles set down in the Declaration of Helsinki and all participants provided written informed consent prior to testing.

### 2.2. Experimental Protocol

All measurements were conducted in one visit. Anthropometric and ACL injury related data were collected first. Then, all participants were given standardized footwear (Adidas Handball Spezial) to avoid differences in shoe properties potentially affecting study outcomes. A five-minute warm-up on an ergometer at a self-selected speed and power was conducted, followed by two submaximal bilateral countermovement jumps (CMJs) and two unilateral CMJs each on each leg.

The participants performed five successful single-leg landings over a 30 cm hurdle in a forward direction onto a force plate (Figure 1). They started in a bilateral stance position at a distance of 40% of their body height to the force platform. They were instructed to jump off both feet, jump over the hurdle, land on one leg in the centre of the force plate, stabilize as quickly as possible, and remain still for eight seconds. For the propulsion phase, they were allowed to use their arms to propel themselves over the hurdle but as soon as they landed and stabilized, they had to place their hands on their hips (Figure 1) [19]. Prior to testing, they had the possibility to try as many jumps as needed to familiarize themselves with the task [19]. The jump-landing was performed alternating between legs with a 30 s rest between each jump [19]. Selection of the first landing leg was randomized. Trials were repeated if the participants touched the hurdle, touched the ground with the opposite leg, re-adjusted their foot position after the initial ground contact, landed on the force plate edge, or failed to place their hands on their hips immediately after landing.

### 2.3. Data Collection and Analysis

Ground reaction force data were collected with a floor-embedded force plate (AMTI, Watertown, MA, USA) at 1000 Hz via Vicon Nexus 2.12.0 software. Additionally, a 2-dimensional high-speed video camera (Casio Exilim EX-F1, Casio Computer Co., Ltd., Tokio, Japan) recorded the testing procedure with 300 fps, to later screen the jumps for potential errors that were missed during data collection (see above).

Data analysis was performed using a custom-written Matlab script (The MathWorks, Version R2019b, Natick, MA, USA). Vertical, medio–lateral, and anterior–posterior ground reaction forces were filtered using a 5th-order zero-lag low-pass Butterworth filter with a cut-off frequency of 50 Hz. The data were cut to the time of initial contact, where the vertical force exceeded 10 N. The TTS (seconds) was defined as the time required by participants to regain a stable position following landing where stability was operationalized as the absence of oscillations in the ground reaction force within certain stability boundaries. The TTS was calculated for each force component. The vertical TTS (TTS-V) was defined as the time required for the vertical force to reach and remain within ±5% of the participant’s body weight for 1 s after landing [20] (Figure 2). Each participant’s body weight was estimated in the first 3 seconds of standing still on the force plate. Medio–lateral TTS (TTS-ML) and anterior–posterior TTS (TTS-AP) were defined as the time required for the ML and AP forces to reach and remain within the overall series mean ± 1 standard deviation for 1 s after landing (Figure 2). A higher TTS represented poorer dynamic postural stability. Means over the best three out of five jump-landing trials for each TTS calculation were used for the statistical analysis.

### 2.4. Statistical Analysis

The statistical analysis was conducted using software IBM SPSS 26.0. (SPSS Inc. Chicago, IL, USA). Descriptive statistics included group means (M) and standard deviations (SDs). Normal distribution was checked using the Shapiro–Wilk test. Demographic and anthropometric data were compared between groups with an independent t-test or, if the data were not normally distributed, with a Mann–Whitney U test.

Hypothesis 1 was tested by comparing the groups without considering the dominance of the injured leg. Given that 15 of the 23 individuals in the ACL group sustained their injury on the non-dominant leg (Table 1), the injured leg of the ACL group was compared to the non-dominant leg of the control group and the contralateral leg of the ACL group was compared to the dominant leg of the control group. Independent t-tests or Mann–Whitney U tests, if the data were not normally distributed, were calculated for the TTS-ML, TTS-AP and TTS-V. The absence of within-subject differences between limbs in the control group in stabilization times were tested with paired t-tests or a Wilcoxon signed-rank test.

Hypothesis 2 was tested by distinguishing between dominant and non-dominant leg ACL injuries. Specifically, we compared the dominant leg of the control group to the dominant leg of the ACL group, which was the injured leg for some individuals and the uninjured contralateral leg for other individuals. Thus, this was an analysis of a between-subject factor with three levels: ACL leg, contralateral leg, control leg. The same procedure was repeated for the non-dominant leg. Non-parametric tests (Kruskal–Wallis) were selected to account for the small sample sizes in the sub-groups when comparing the TTS-ML, TTS-AP and TTS-V between ACL legs, contralateral legs, and control legs. Dunn–Bonferroni post hoc tests were applied to investigate pairwise comparisons. We rearranged the data in figures based on the dominance of the legs, with the aim of visualizing and comparing the effects of leg dominance regarding dynamic postural stability.

The significance level was set at alpha = 0.05. Effect sizes Cohen’s d and Rosenthal’s r [21] were calculated for parametric and non-parametric statistical comparisons. Cohen´s d was interpreted as follows: d = 0.20 for a small, d = 0.50 for a moderate and d = 0.80 for a large effect size and Rosenthal´s r was interpreted as follows: r = 0.10 for a small, r = 0.30 for a moderate and r = 0.50 for a large effect size.

## 3. Results

Demographic characteristics showed no significant differences between the injured and the control group (Table 1). Following screening of the high-speed video footage, the results of some trials had to be excluded, since retrospectively it was noticed that the landing had not been performed according to the instructions. This resulted in the following sample sizes: *n* = 23 individuals with a previous ACL injury, of whom 8 had injured their dominant leg and 15 their non-dominant leg. When analysing the contralateral leg, *n* = 22 participants could be included in the ACL group, of whom 14 were the dominant legs and 8 the non-dominant legs. In the control group, we had *n* = 18 valid datasets for both the dominant and non-dominant leg landings.
brainsci-13-01721-t001_Table 1Table 1Participant information for the ACL group and the control group.
ACL Group(*n* = 23 Females)Control Group(*n* = 20 Females)*p*-Value *Age (years)24.3 ± 3.224.8 ± 2.10.237Height (m)1.7 ± 0.11.7 ± 0.00.566Weight (kg)59.2 ± 5.061.0 ± 5.70.451BMI (kg/m^2^)21.0 ± 2.421.9 ± 1.60.295ACLRYes: 21No: 2

Leg dominance Right: 23 Right: 19; left: 1 
Injury side Dominant leg: 8 Non-dominant leg: 15 n/a 
Time since injury (year)5.0 ± 2.6 n/a
Reinjury on ipsilateral side 3/23n/a
Graft typeBPTB: 2ST: 16QT: 5n/a
Physical activity (day/week)4.4 ± 1.54.9 ± 1.20.207Physical activity (minutes/session)91.3 ± 36.799.5 ± 39.40.532IKDC (%) 84.3 ± 4.7n/a
Data are presented as mean (±standard deviation). ACLR: Anterior cruciate ligament reconstruction. Leg dominance: Preferred kicking leg. ST: Semitendinosus-gracilis tendon graft, BPTB: Bone–patellar tendon–bone graft, QT: Quadricep tendon graft. Two participants had a re-injury on the ipsilateral leg, resulting in 23 graft types. IKDC: International Knee Documentation Committee. n/a: not applicable; * *p*-values considered significant at *p* < 0.05.

### 3.1. Dynamic Postural Stability Irrespective of Leg Dominance

In the medio–lateral direction, the TTS was significantly reduced for the ACL injured leg compared to the non-dominant leg of the control group with a moderate effect size (*p* = 0.006, d = 0.91). On average, the ACL group needed 0.33 ± 0.11 s longer to stabilize on their injured legs compared to the control group (Figure 3, top row). Further, the ACL group showed an increased TTS-ML on their contralateral leg compared to the dominant leg of the control group with a moderate effect size; however, this comparison did not reach statistical significance (*p* = 0.050, d = 0.64) (Figure 3, top row). The TTS-V was significantly higher in the ACL injured leg compared to the non-dominant leg of the control group with a moderate effect size (*p* = 0.030, r = 0.34) (Figure 3, middle row). Further, the ACL group demonstrated an increased TTS-V on their contralateral leg compared to the dominant leg of the control group with a small effect size; however, this comparison did not reach statistical significance (*p* = 0.106, r = 0.26) (Figure 3, middle row). No significant differences and negligible effect sizes were found for the comparisons of the TTS-AP between the ACL leg and the non-dominant leg of the control group (*p* = 0.741, r = 0.05), or between the contralateral leg and the dominant leg of the control group (*p* = 0.522, r = 0.08) (Figure 3, bottom row).

### 3.2. Effect of ACL Injury on Dynamic Postural Stability When Considering Leg Dominance

When analyzing dominant legs (i.e., the dominant leg of individuals with dominant-leg ACL injuries vs. the dominant (uninjured) leg of individuals with non-dominant leg injuries vs. the dominant leg of the control group), there was a significant effect of the “group” on the TTS-ML (*p* = 0.007, r = 0.46, Figure 4, top left). Post hoc tests indicated that the ACL group took 0.59 ± 0.15 s longer to stabilize on their ACL-injured dominant leg compared to the dominant leg of the control group (*p* = 0.006, r = 0.49).

When analyzing non-dominant legs (i.e., the non-dominant leg of individuals with non-dominant-leg ACL injuries vs. the non-dominant (healthy) leg of individuals with dominant leg injuries vs. the non-dominant leg of the control group), there was a significant effect of the ”group” on the TTS-ML (*p* = 0.036, r = 0.35, Figure 4, top right). Post hoc tests indicated that the ACL group took 0.47 ± 0.17 s longer to stabilize on their healthy non-dominant leg compared to the non-dominant leg of the control group (*p* = 0.034, r = 0.40).

In contrast, no significant post hoc comparisons were observed between the control group and the ACL group if the injury had occurred on the non-dominant leg (blue vs. white boxes in Figure 4, dominant contralateral leg vs. dominant leg of the control group: *p* = 1.000, r = 0.11; non-dominant ACL leg vs. non-dominant leg of the control group: *p* = 0.497, r = 0.24). In other words, the TTS-ML was significantly increased compared to the control legs but only for the ACL subgroup who had injured their dominant leg and, in fact, this observation was made for the previously injured leg and the healthy contralateral leg (red vs. white boxed in Figure 4).

Finally, there were no significant differences between groups in the TTS for the vertical (Figure 4, middle row: dominant leg: *p* = 0.263; non-dominant: *p* = 0.158) and the anterior–posterior directions (Figure 4, bottom row: dominant leg: *p* = 0.838; non-dominant: *p* = 0.602).

## 4. Discussion

This study examined dynamic postural stability in a unilateral jump-landing task, measured through the TTS, in physically active females following a unilateral ACL injury. As hypothesized, we found significant reductions in dynamic postural stability for the ACL injured leg. Further, we had hypothesized that after an ACL injury the contralateral leg would also show poorer dynamic stability than healthy knees. And indeed, we observed elevated medio–lateral TTS durations (Figure 3); however, statistically we could only confirm a trend (*p* = 0.050) and therefore have to classify this result as inconclusive. Interestingly, when distinguishing whether the ACL injury had occurred on the dominant or the non-dominant side, we found poor dynamic postural stability on both legs (injured and contralateral compared to healthy) if the injury had occurred on the dominant leg. These results were significant despite a small sample size. In contrast, when the non-dominant leg had suffered an ACL injury, dynamic postural stability on both legs did not significantly differ from the control group.

### 4.1. Dynamic Postural Stability Irrespective of Injury Side

Our observation that ACL-injured knees need a longer TTS in a dynamic landing task is in line with some previous studies [12], but differs from others [22], or was only found in diagonal but not in linear forward jumps [23]. One potential reason for the longer TTS on the ACL-injured leg could be deficits in mechanoreceptor function in the reconstructed ACL [24] that likely persist despite the fact that some reinnervation is known to occur after ACL reconstruction [25].

Our results regarding stabilization on the contralateral leg are inconclusive and call for additional research. Such research seems warranted considering that previous studies reported bilateral proprioceptive deficits in individuals with unilateral ACL injuries [10,11]. In particular, Culvenor et al. [10] identified larger proprioceptive deficits in the medio–lateral direction for both legs during a static squat task. A previously proposed explanation for these findings could be the disruption of afferent feedback to the brain, leading to a reorganization of motor areas in the CNS [4,5]. However, our observation that TTS results differed substantially if the injury had occurred on the dominant leg as opposed to the non-dominant side suggests that pooling outcomes from both injury sides might be inappropriate. Such pooling might on the one hand wash out effects that are present in one sub-group and, on the other hand, falsely suggest the presence of effects in other subgroups. It is a possibility that the inconsistent post ACLR TTS results in the literature may stem from inconsistent injury sides within the patient groups.

### 4.2. Effect of Leg Dominance on Dynamic Postural Stability

Limb dominance arises due to the specialization of the brain hemispheres for specific tasks or functions [8,26]. In the lower limbs, as opposed to the upper limbs where sidedness often implies a substantial difference in performance (e.g., in handwriting, throwing, etc.), outcome variables or performance in motor tasks often do not differ between the dominant and non-dominant leg [27]. One explanation for this observation is that most motor tasks involving the lower extremities are bipedal; hence, both legs receive a similar amount of training. Our finding that stabilization in the control group did not differ between limbs (Figure 3, third column) is hence in line with most scientific literature [27]. However, similar performance outcomes may be reached by different motor coordination patterns, which Promsri and colleagues [18] reported for various unipedal balance exercises [18]. Brain hemispheric specialization seems to produce differences in how afferent information is weighted and processed [8,26] and may lead to the reported tendency to rely more on the preferred limb [28]. Consequences of limb dominance could be, on the one hand, the unequal likelihood of injuring the dominant versus the non-dominant leg, as reported by several studies [9]. On the other hand, it could also imply, as our results suggest, that the magnitude of disturbance to the sensorimotor control caused by the injury may depend on the side of the injury. While, to the best of our knowledge, our study is the first to explicitly distinguish the injury side in the analysis of post-ACL stabilization experiments, several previous studies showed that unilateral injury can cause bilateral neuromuscular control deficits [10,11]. Furthermore, studies on motor deficits in patients who had suffered an unilateral stroke found not only deficits in the contralateral and ipsilateral limbs, but similar to our study found that motor deficits depended on hemispheric specialization [29].

There are some limitations to this study. First, only female participants were included; hence, our findings cannot be generalized to male individuals. In fact, since some studies report differences in injury sides between the sexes [9], the possibility that the factors sex and limb dominance may interact in their effects on post-ACL stabilization outcomes can currently not be ruled out. Second, the sample size for participants with a dominant leg injury was small with only eight participants recruited. While the observed effect sizes were large enough to yield significant main effects, some of the post hoc comparisons may have been underpowered. Based on our post hoc sensitivity analysis (Section 2.1), the current study reached a power of 0.8 only for effect sizes larger than 0.77. Third, our study design cannot reveal whether the neuromuscular deficits observed for the ACL group in this study were pre-existing or a result of the injury and subsequent reconstruction. Fourth, we did not control for severity of the injury, which may differ between the ACL sub-groups. Fifth, it should be noted that the TTS only measures overall postural stability and cannot be attributed to control deficits in the knee alone. Sixth, the influence of lower limb strength and interlimb differences in strength on dynamic stability outcomes was not explicitly addressed in this study. Pre-existing differences in lower limb strength could potentially bias the results, although, contrary to the effects of limb dominance on arm strength, the difference in leg strength due to leg dominance are often small and an unlikely injury factor [30]. Seventh, an analysis of the effects of graft types was not conducted in the current study. Lastly, we did not collect data on the implemented rehabilitation program. Graft type and the rehabilitation program may impact dynamic postural stability.

### 4.3. Practical Implications

As already pointed out, if the results of the current study are corroborated in future research, then post-ACL-injury assessments of balance and stabilization should not pool data from patients who suffered their injury on different sites. This would also suggest that in return-to-sport assessments and in the planning of rehabilitative measures, the injury side and its match with leg dominance should be considered. Particularly if the injury occurred on the dominant leg, our results suggest that physicians, physio and training therapists should expect poorer stabilization results on both legs. And they should therefore also enhance their focus on contralateral leg training in ACL injury rehabilitation programs.

Our results also suggest that stability assessments interpreting asymmetries between an injured and a “healthy” contralateral leg should be complemented with comparisons to reference data from uninjured volunteers, as bilateral deficits in sensorimotor control functions need to be taken into consideration.

## 5. Conclusions

The current study hypothesized and found significant reductions in dynamic postural stability for the ACL-injured leg approximately 5 years post-injury. Further, we had hypothesized poorer dynamic stability also in the contralateral leg, for which we could only confirm a statistical trend. However, our results suggest that when the ACL injury occurs on the dominant leg, poorer dynamic postural stability is found on both legs (injured and contralateral). In contrast, when the non-dominant leg suffers an ACL injury, dynamic postural stability on both legs differs only marginally from the control group.

## Figures and Tables

**Figure 1 brainsci-13-01721-f001:**
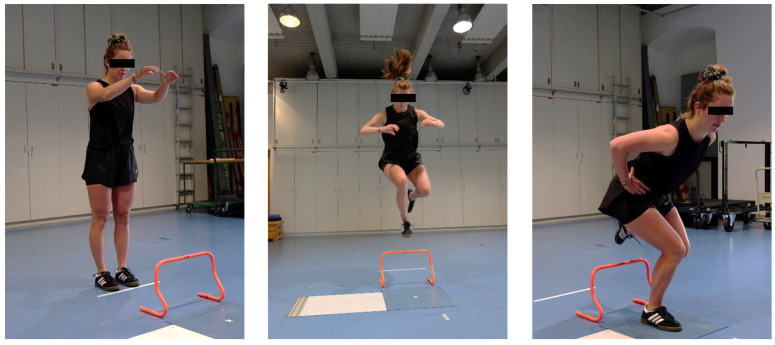
Unilateral jump landing task over 30 cm hurdle onto force plate (blue).

**Figure 2 brainsci-13-01721-f002:**
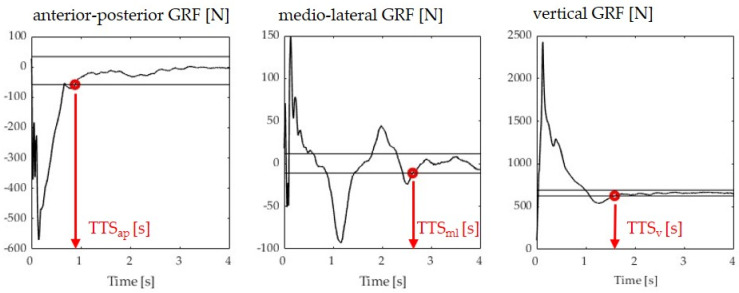
Sample images illustrating the TTS determination in the three anatomical directions. Black horizontal lines define the boundaries, as explained in the text. Red dots define the time point where stability is achieved (last time point a boundary is crossed).

**Figure 3 brainsci-13-01721-f003:**
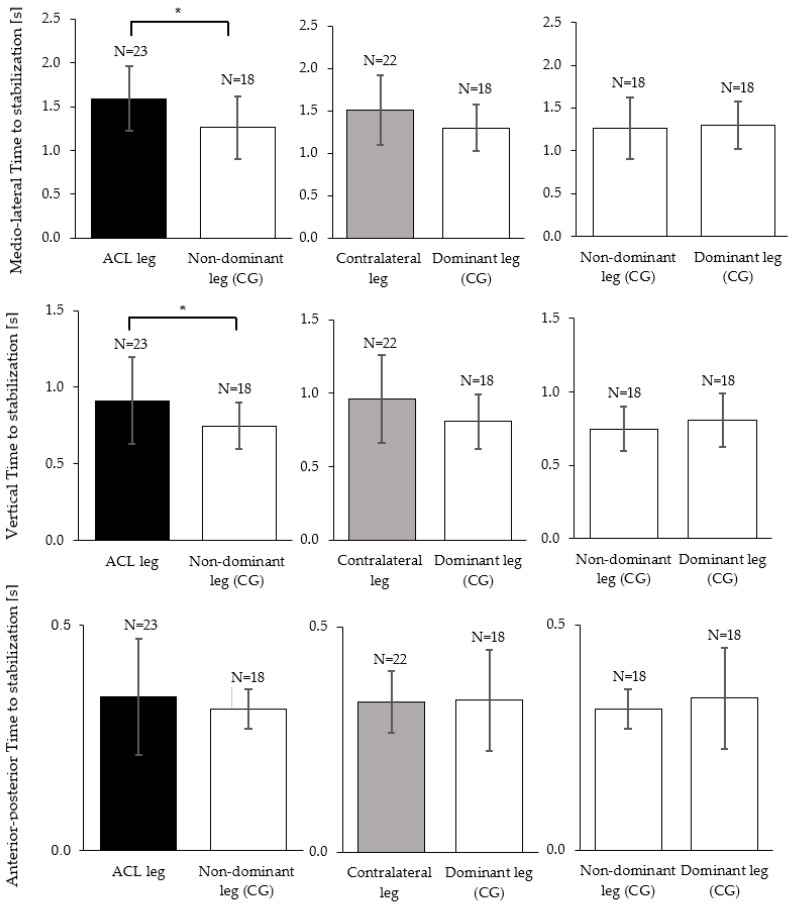
Mean and standard deviation of the time to stabilization (TTS-ML, TTS-AP, TTS-V) between ACL injured (black), contralateral (grey) and the control group legs (white). In TTS-ML, TTS-AP independent sample t-tests/Mann–Whitney U tests found statistically significant differences as marked by the asterisks (* *p* < 0.05). Within the control group (right column), paired t-tests found no differences between the dominant and the non-dominant leg.

**Figure 4 brainsci-13-01721-f004:**
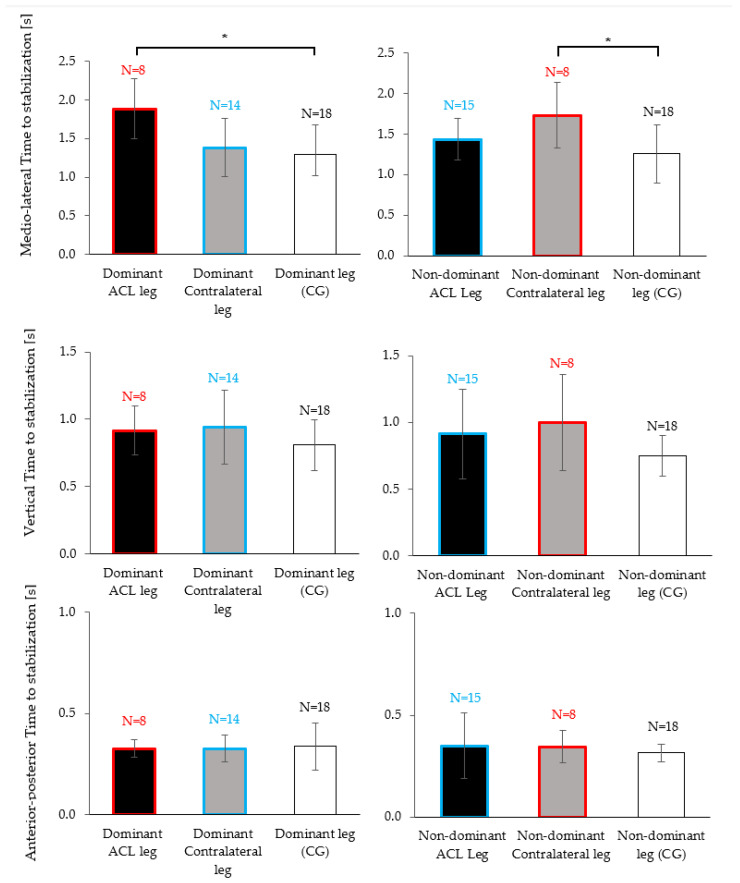
Mean and standard deviation of the time to stabilization (TTS-ML, TTS-AP, TTS-V) between ACL injured (black), contralateral (grey) and the control group legs (white). Red boxes correspond to the TTS of individuals with dominant leg injuries (either for the ACL or contralateral leg) and blue boxes correspond to the TTS of individuals with non-dominant leg injuries (either for the ACL or contralateral leg). Each graph corresponds to a Kruskal–Wallis test followed by a Dunn–Bonferroni post hoc analysis if it found differences. Significant pairwise differences are indicated with asterisks (* *p* < 0.05).

## Data Availability

The data presented in this study are available on request from the corresponding author. The data are not publicly available due to privacy concerns.

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
