# Peer review of "Bilateral Deficits in Dynamic Postural Stability in Females Persist Years after Unilateral ACL Injury and Are Modulated by the Match between Injury Side and Leg Dominance"

_brainsci, 2023, doi:10.3390/brainsci13121721_

Round 1

Reviewer 1 Report

Comments and Suggestions for Authors

Dear Authors,

in my opinion the manuscript is original and interesting because the analysis focused on underlying any influence that an injury would bring to the uninjured side, and the importance that dominance would play in postural stability, providing a new point of view when evaluating the return-to-play of an injured athlete.

However, I have some concerns about the methodological implant of the study and some critical issues should be addressed.

Major revisions

Whole manuscript: a careful English revision is necessary because readability should be improved. For example: “to the already injured ipsilateral” (page 1, line 29) could be changed to “as a recurrence”; “reconstructed ipsilateral ACL” (page 1, line 31) could be changed to “reconstructed ACL”, etc.

Methods: please, report included patients in the Results section. Accordingly, Table 1 should be reported in the “Results” section.

Methods: please, clarify how the dominant leg was defined.

Methods: lower limbs strength, and any intrapersonal difference between dominant and non-dominant leg, might influence dynamic stability outcome, and any pre-existing difference could bias the results. The sample characterization might be significantly improved including these data. On the other hand, if this information is not obtainable, it should be discussed in the Limitations Subsection. At the same time, graft type and inclusion of different grafts could play a role in functional outcome, as well as rehabilitation program post injury; they should be discussed as a limitation.

Results: patients assessed for eligibility and patients excluded should be clarified in the results section, characterizing at least the main cause of exclusions. Moreover, consider providing a flow diagram for included and excluded patients.

Discussion – Practical implication (lines 385-388): this subsection should be improved by mentioning both preventive and rehabilitative intervention on balance control in athletes. According to this, you should cite the following references:

·      Lippi L, et al. Effects of blood flow restriction on spine postural control using a robotic platform: A pilot randomized cross-over study. J Back Musculoskelet Rehabil. Published online August 30, 2023. doi:10.3233/BMR-230063

·      Hötting K, et al. Balance Expertise Is Associated with Superior Spatial Perspective-Taking Skills. Brain Sci. 2021;11(11):1401. Published 2021 Oct 24. doi:10.3390/brainsci11111401

Minor revisions

Title: please, the title should be more specific (e.g., clarifying it’s on a female population).

Introduction: please, improve the incipit of the paragraph.

Introduction (page 2, line79): please, since not all the injured patients that were included underwent reconstruction, you should correct “ACLR” with “ACL”.

Table 1: in the ACL group, 21 underwent ACLR, and 2 did not. Though, if we sum up the graft type (BPTB: 2, ST: 16, and QT: 5), it adds up to 23. Does this imply multiple grafts in some patients? Please, clarify.

Author Response

Dear Reviewers, dear Editors

thank you for your time and for providing valuable feedback on our manuscript! We revised the paper and incorporated your comments as detailed below. All changes are highlighted in red in the manuscript.

In our point-by-point response, the reviewer comments are printed in bold. Our responses in italics. Revised texts from the manuscript we included in red.  

Thank you for your help with improving the manuscript,

in the name of all authors,

sincerely,

Peter Federolf

Reviewer 1

Dear Authors,

in my opinion the manuscript is original and interesting because the analysis focused on underlying any influence that an injury would bring to the uninjured side, and the importance that dominance would play in postural stability, providing a new point of view when evaluating the return-to-play of an injured athlete.

Response:

We would like to thank you for your positive assessment and your valuable feedback.

Major revision:

Whole manuscript: a careful English revision is necessary because readability should be improved. For example: “to the already injured ipsilateral” (page 1, line 29) could be changed to “as a recurrence”; “reconstructed ipsilateral ACL” (page 1, line 31) could be changed to “reconstructed ACL”, etc.

Response:

Thank you for your suggestions. We added the suggestions in the manuscript:

“More than 20% of individuals who underwent primary anterior cruciate ligament (ACL) reconstruction (ACLR) will suffer a secondary ACL injury, either as a recurrence to the already injured ipsilateral or to the contralateral leg [1]. In fact, the risk of sustaining a new rupture to the healthy contralateral ACL has been reported to be significantly higher than the risk of a re-rupture of the reconstructed ACL [2].”
(page 1, line 27-32)

We also re-read the manuscript and improved the readability where we recognized similar issues. Unfortunately, sometimes it was our impression that in order to be precise, complex wording was in some instances not avoidable.

Methods: please, report included patients in the Results section. Accordingly, Table 1 should be reported in the “Results” section.

Response:

As suggested, Table 1 was relocated to the result section (page 6, line 211-216)

Methods: please, clarify how the dominant leg was definded

Response:

Thank you for this comment, this definition was indeed missing. Leg dominance was determined as the preferred kicking leg. We added this information within the footnote of the demographics table (page 3, line 98-99)

Methods: lower limbs strength, and any intrapersonal difference between dominant and non-dominant leg, might influence dynamic stability outcome, and any pre-existing difference could bias the results. The sample characterization might be significantly improved including these data. On the other hand, if this information is not obtainable, it should be discussed in the Limitations Subsection. At the same time, graft type and inclusion of different grafts could play a role in functional outcome, as well as rehabilitation program post injury; they should be discussed as a limitation.

Response:

Thank you. We added these points in the limitation section.

“Second, the sample size for participants with a dominant leg injury was small with only 8 participants recruited. While the observed effect sizes were large enough to yield significant main effects, some of the post-hoc comparisons may have been underpowered. Based on our post-hoc sensitivity analysis (section 2.1) the current study reached a power of 0.8 only for effect sizes larger than 0.77. Third, our study design cannot reveal whether the neuromuscular deficits observed for the ACL group in this study were pre-existing or a result of the injury and subsequent reconstruction. Fourth, we did not control for severity of the injury, which may differ between the ACL sub-groups. Fifth, it should be noted that the TTS only measure overall postural stability and cannot be attributed to control deficits in the knee alone. Sixth, the influence of lower limb strength and interlimb differences in strength on dynamic stability outcomes was not explicitly addressed in this study. Pre-existing differences in lower limb strength could potentially bias the results, although, contrary to effects of limb dominance on arm strength, difference in leg strength due to leg dominance are often small and an unlikely injury factor [30]. Seventh, an analysis of the effects of graft types was not conducted in the current study. Lastly, we did not collect data on the implemented rehabilitation program. Graft type and the rehabilitation program might impact dynamic postural stability.”
(Page 11, line 394-411)

Results: patients assessed for eligibility and patients excluded should be clarified in the results section, characterizing at least the main cause of exclusions. Moreover, consider providing a flow diagram for included and excluded patients.

Response:

We revised the section explaining the exclusion procedure and simplified the wording. This section now reads as follows:

“Following screening of the high-speed video footage, the results of some trials had to be excluded, since retrospectively it was noticed that the landing hadn’t been performed according to instruction. This resulted in the following sample sizes: n = 23 individuals with a previous ACL injury, of which 8 had injured their dominant leg and 15 their non-dominant leg. When analysing the contralateral leg, n = 22 participants could be included in the ACL group, of which 14 were the dominant legs and 8 the non-dominant legs. In the control group we had n = 18 valid datasets for both the dominant and non-dominant leg landings.”

(page 6, line 202-209)

Considering that the exclusion was based mainly on technical issues, we felt that a flow diagram was not needed an would overemphasize a relatively simple process.

Discussion – Practical implication (lines 385-388): this subsection should be improved by mentioning both preventive and rehabilitative intervention on balance control in athletes. According to this, you should cite the following references:

  • Lippi L, et al. Effects of blood flow restriction on spine postural control using a robotic platform: A pilot randomized cross-over study. J Back Musculoskelet Rehabil. Published online August 30, 2023. doi:10.3233/BMR-230063
  • Hötting K, et al. Balance Expertise Is Associated with Superior Spatial Perspective-Taking Skills. Brain Sci. 2021;11(11):1401. Published 2021 Oct 24. doi:10.3390/brainsci11111401

Response:

We have improved the section regarding practical implications.

Regarding the two suggested citations, we would like to ask whether a copy-past error may have happened? These two papers are quite far off-topic, as they have nothing to do with ACL injuries or time-to-stabilization. We don’t see where reasonably we could refer to them in the context of our paper.  

Minor revision:

Title: please, the title should be more specific (e.g., clarifying it’s on a female population).

Response:

We formulated a more specific title: “Bilateral deficits in dynamic postural stability in females persist years after unilateral ACL injury and are modulated by the match between injury side and leg dominance“

Introduction: please, improve the incipit of the paragraph.

Response:

We reworded the beginning of the first paragraph. We agree that this could have been more concise. 

Introduction (page 2, line79): please, since not all the injured patients that were included underwent reconstruction, you should correct “ACLR” with “ACL”.

Response:

Agreed. Thank you. We reworded this to:

 “Therefore, the objective of this study was to assess neuromuscular control deficits following unilateral ACL injury through TTS during single-leg jump-landing tasks on both legs and in comparison, to a control group.” (page 2, line 79-81)

Table 1: in the ACL group, 21 underwent ACLR, and 2 did not. Though, if we sum up the graft type (BPTB: 2, ST: 16, and QT: 5), it adds up to 23. Does this imply multiple grafts in some patients? Please, clarify.

Response:

Yes: two participants had a re-injury on the ipsilateral leg, resulting in 23 graft types. We added this information within the footnote of the demographics table.  (page 6, line 213-216)

Reviewer 2 Report

Comments and Suggestions for Authors

Thank you for the opportunity to review your manuscript, “Bilateral deficits in dynamic postural stability persist years after unilateral ACL injury and are modulated by the match between injury side and leg dominance.”

The article is well-written, methodologically sound and relatively easy to follow. There are a few minor issues that would enhance the manuscript.

The sample size calculation should be added, or a mention of the statistical power of the results.

The conclusions section should be reworded with more conclusive wording. It is currently worded as if it were the results section.

Author Response

Dear Reviewers, dear Editors

thank you for your time and for providing valuable feedback on our manuscript! We revised the paper and incorporated your comments as detailed below. All changes are highlighted in red in the manuscript.

In our point-by-point response, the reviewer comments are printed in bold. Our responses in italics. Revised texts from the manuscript we included in red.  

Thank you for your help with improving the manuscript,  

in the name of all authors,

sincerely,

Peter Federolf

Reviewer 2

Thank you for the opportunity to review your manuscript, “Bilateral deficits in dynamic postural stability persist years after unilateral ACL injury and are modulated by the match between injury side and leg dominance.”

The article is well-written, methodologically sound and relatively easy to follow. There are a few minor issues that would enhance the manuscript.

Response:

We would like to thank you for your positive assessment and your valuable feedback.

The sample size calculation should be added, or a mention of the statistical power of the results.

Response:

Thank you for this valid remark. We did not perform an a-priori sample size calculation. However, a post-hoc sensitivity analysis is now included in the manuscript: 

“A post-hoc sensitivity analysis with a significance level of alpha = 0.05 indicated that this sample size enabled us to detect between-group effects (ACL vs. control group) of moderate to large effect sizes (Cohen’s d > 0.77) with a desired power of 0.8.”
(page 2-3, line 94-97)

The conclusions section should be reworded with more conclusive wording. It is currently worded as if it were the results section.

Response:

Agreed. We reworded this section to better reflect the conclusive nature of this chapter.

Reviewer 3 Report

Comments and Suggestions for Authors

The objective of the paper "Bilateral deficits in dynamic postural stability persist years after unilateral ACL injury and are modulated by the match between injury side and leg dominance" was to evaluate neuromuscular control deficits following unilateral ACLR through TTS during single-leg jump-landing tasks on both legs, comparing them to a control group. In addition to the vertical component of the ground reaction force, the authors specifically focused on the anterior-posterior and mediolateral force components, as knee injury risk factors are often associated with mediolateral instability. The study aimed to test the primary hypothesis (H1) that dynamic postural stability, quantified by TTS, would be lower for both the healthy contralateral leg and the ACL-injured leg than a control group with no history of knee injury. Furthermore, they explored the secondary hypothesis (H2) that the match between leg dominance and the side of the injury would influence dynamic postural stability in the ACL group.

In my opinion, the paper is well-written. However, clarification is needed regarding the authors' definition of a "stable position" (line 144). Additionally, it would be beneficial to include a sample figure illustrating the movement sequence with a graph depicting the three components of the ground reaction forces, marking the moment the authors consider the test subject to have achieved a stable position.

There is a minor error regarding punctuation; there should not be a period after the title, which the editors will likely correct.

Author Response

Dear Reviewers, dear Editors

thank you for your time and for providing valuable feedback on our manuscript! We revised the paper and incorporated your comments as detailed below. All changes are highlighted in red in the manuscript.

In our point-by-point response, the reviewer comments are printed in bold. Our responses in italics. Revised texts from the manuscript we included in red.  

Thank you for your help with improving the manuscript,

in the name of all authors,

sincerely,

Peter Federolf

Reviewer 3

The objective of the paper "Bilateral deficits in dynamic postural stability persist years after unilateral ACL injury and are modulated by the match between injury side and leg dominance" was to evaluate neuromuscular control deficits following unilateral ACLR through TTS during single-leg jump-landing tasks on both legs, comparing them to a control group. In addition to the vertical component of the ground reaction force, the authors specifically focused on the anterior-posterior and mediolateral force components, as knee injury risk factors are often associated with mediolateral instability. The study aimed to test the primary hypothesis (H1) that dynamic postural stability, quantified by TTS, would be lower for both the healthy contralateral leg and the ACL-injured leg than a control group with no history of knee injury. Furthermore, they explored the secondary hypothesis (H2) that the match between leg dominance and the side of the injury would influence dynamic postural stability in the ACL group.

In my opinion, the paper is well-written.

Response:

We would like to thank you for your positive assessment and your valuable feedback.

However, clarification is needed regarding the authors' definition of a "stable position" (line 144).

Response:

Agreed. We reworded this description:

 “TTS (seconds) was defined as the time required by participants to regain a stable position following landing where stability was operationalized as the absence of oscillations in the ground reaction force within certain stability boundaries.”
(page 4, line 149-151)

Additionally, it would be beneficial to include a sample figure illustrating the movement sequence with a graph depicting the three components of the ground reaction forces, marking the moment the authors consider the test subject to have achieved a stable position.

Response:

We added three figures depicting the three ground reaction force components and the location where a stable position is achieved (red dots). The figure can be found in the method section on page 5, line 161-166

There is a minor error regarding punctuation; there should not be a period after the title, which the editors will likely correct.

Response:

This has been changed: “Bilateral deficits in dynamic postural stability in females persist years after unilateral ACL injury and are modulated by the match be-tween injury side and leg dominance”
